# Does Income Inequality Explain the Geography of Residential Burglaries? The Case of Belo Horizonte, Brazil

Rafael G. Ramos 

Department of Geography, University of California Santa Barbara, Santa Barbara, CA 93106, USA; rafaelramos@ucsb.edu

**Abstract:** The relationship between crime and income inequality is a complex and controversial issue. While there is some consensus that a relationship exists, the nature of it is still the subject of much debate. In this paper, this relationship is investigated in the context of urban geography and whether income inequality can explain the geography of crime within cities. This question is examined for the specific case of residential burglaries in the city of Belo Horizonte, Brazil, where I tested how much burglary rates are affected by local average household income and by local exposure to poverty, while I controlled for other variables relevant to criminological theory, such as land-use type, density and accessibility. Different scales were considered for testing the effect of exposure to poverty. This study reveals that, in Belo Horizonte, the rate of burglaries per single family house is significantly and positively related to income level, but a higher exposure to poverty has no significant independent effect on these rates at any scale tested. The rate of burglaries per apartment, on the other hand, is not significantly affected by either average household income or exposure to poverty. These results seem consistent with a description where burglaries follow a geographical distribution based on opportunity, rather than being a product of localized income disparity and higher exposure between different economic groups.

**Keywords:** crime; burglary; mapping; GIS; spatial analysis; urban geography; human geography

## 1. Introduction

The relationship between crime and income inequality is complex and controversial. While there is some consensus that a relationship exists, the nature of it is the subject of much debate, which is further intensified by its relevance to the public debate [1–5]. In this paper, this relation is investigated in the context of urban geography: Does income inequality explain the geography of crime within cities?

This question has been relatively overlooked, and a satisfying answer is still absent. Traditionally, the theoretical base for a relationship between income inequality and crime has been given by theories such as strain theory [6], relative deprivation theory [7] and the economic theory of crime [8,9]; however, none of these theories explicitly address the spatial aspect. Conversely, income inequality is not prominently featured in any of the notable spatial theories of crime such as routine activities theory [10] and crime pattern theory [11,12]. While a potential connection between income inequality and the spatial distribution of crime within a city could be conjectured by combining these different theories, this has not often been tested. Most empirical studies comparing crime rates and income inequality have been conducted at more macro-scales such as comparing cities, counties, or even whole regions or countries [1,3,13–19]; contrasting with that, I found only five studies analyzing this relationship at a within-city scale [20–24].

This research proposes a model to test whether and how income and its unequal distribution can explain the geography of residential burglaries within a city, while taking other factors potentially

relevant to crime into account. More specifically, the questions aimed to be answered by the model are the following three:

1.　What is the effect of local average income level on local burglary risk?
2.　What is the effect of higher exposure to poverty on burglaries, when the local average income level is controlled?
3.　What is the scale at which this exposure is most relevant?

In other words, this research aims to answer questions such as: Do higher income locations feature higher burglary risk? Given a higher income location, will the presence of nearby surrounding poverty increase the risk of burglary at that higher income location? What is meant by nearby, and at what scale is the effect of proximity to poverty relevant, if any?

The model was tested for residential burglaries in the city of Belo Horizonte, Brazil. For the case tested in this paper, a strong positive correlation was found between burglary risk at houses and local average income, with the model explaining 61% of the variance. Exposure to poverty was found to be insignificant in independently influencing burglary risk at single family houses, regardless of the scale considered. For the risk of burglary at apartments, on the other hand, no significant effect was found for either local average income or exposure to poverty.

Burglary was chosen for this study since, as a type of property crime, it potentially has a more straightforward connection to income and related matters. Furthermore, since the targets of burglary are both explicit in space and unmoving (i.e., the residences are fixed in location), examining the spatial distribution is a less ambiguous task.

The study area for this research was the city of Belo Horizonte, Brazil, the third largest city of the country in terms of metropolitan population. For the purposes of studying the relationship between crime and income inequality, Brazil is an important case, since it not only the ranks among the highest in violent crime [25], it also ranks among the highest in income inequality [26].

Within Brazil, Belo Horizonte is an adequate choice of city. Crime is more concentrated in Brazil's largest cities, such as Belo Horizonte—Brazil's third largest metropolitan area. Moreover, despite originally being a planned city, Belo Horizonte's urban structure better matches that of other larger Brazilian cities in marked contrast to Rio de Janeiro and its unique landscape, the formal urban design of Brasília, or the sheer magnitude of São Paulo. That, combined with the fact that Belo Horizonte's demographic profile closely matches the national average, should facilitate future comparisons to other cities in Brazil.

This article is organized as follows. The remainder of this section discusses the key theories and studies related to the topic of crime, income inequality and their geographies, which were used as the theoretical base for the model proposed here. Section 2 (Material and Methods) describes the model proposed, as well as the data used in this study. Section 3 (Results) summarizes the main results obtained, with Section 4 (Discussion) providing an interpretation of these results, a comparison between this study and previous ones (including methodological aspects), and a summary of the main conclusions.

*Theoretical Background and Literature Review*

The theoretical base for a connection between income inequality and crime has been explored by different theoretical lines. In strain theory [6] (and the similar relative deprivation theory [7]), the presence of social inequalities (income inequality included) results in higher feelings and perceptions of injustice in society, leading to psychological strain that culminate in crime. Within this framework, the poor, confronted with what is perceived as the unattainable success standards of the middle-class, more likely suffer from psychological strain and resort to crime as an alternative route to success. The economic theory of crime [8,9], on the other hand, states that income inequality boosts crime because it increases the number of economically strained individuals who are be more pressured to resort to crime as a money-earning option while also sustaining a number of wealthy individuals

who are adequate targets of property theft. Contrasting with the emotion-drive explanation of the strain and relative deprivation theories, the connection between crime and income inequality is instead economical, assuming rational decision making. The final general effect, though, is similar. However, while these theories offer plausible explanations for a connection between crime and income inequality, how this is manifested in space has not been directly analyzed.

Theories that do provide an explanation for spatial patterns crime, on the other hand, usually do not feature income-related aspects as prominent factors, and the specific connection between income inequality and the geography of crime is again not clear in these theories. Routine activities theory [10] is one of the more prominent spatial theories of crime. As remarked by routine activities theory, in order for a crime to happen, three elements must converge in time and space: The presence of both a motivated offender and a suitable target, as well as the absence of a capable guardian. Based on this premise, routine activities theory then analyzes crime as a function of how these three different elements may vary in time and across space. Burglaries, for instance, are more common during the day because residences are more often vacant during this time, leading to increased opportunities for burglaries. Income is included in this framework as a modulator of opportunities and potential gains, with the theory explaining how times of economic boon could feature an increase in property crime rates: Increased wealth and income not only increase the consumption and acquisition of goods such as expensive electronics (increase in values targets) but also increase how often people go out at night for leisure activities (leaving not only their residences vacant but also increasing their own exposure to crime).

Crime pattern theory [11,12] further refined the approach from routine activities theory, presenting a more explicit spatial framework to explain crime in terms of flows of offenders, targets and guardians. According to crime pattern theory (and following routine activities theory), spatial patterns of crime are a product of the combined activities spaces of offenders, targets and guardians—an activity space being the collection of places and regions most frequently visited by an individual based on their daily routine. Activities spaces, on the other hand, are mostly defined and constrained by a few key nodes and connecting edges, such as a person's home location, work-place, and preferred leisure venues (as well as the routes connecting these places). Therefore, spatial patterns of crime can be understood by analyzing these key nodes, an approach that has been used by the police to identify the likely home location of serial offenders.

Other important criminological theories can also be cited as relevant to the topic of crime, income inequality, and their geographies (even if these connections are not totally fleshed out). Rational choice theory [27] models offender behavior as multi-level rational decision-making processes in which considerations of proximity and accessibility play a role in target selection. The income level of the target can be considered relevant within this framework, determining potential gain and thus influencing the target selection process. More indirectly, income and socioeconomic status can be considered as determining an offender's background and social connections, factors which are at least partially contemplated in the theory. However, a straightforward, explicit model relating income and income inequality to the spatial distribution of crime has not been provided. Finally, social disorganization theory [28] (and related theories such as collective efficacy [29]) should be listed as an important theory that features both income and spatial aspects to a limited extent. The spatial variation of crime is explained as a function of neighborhood characteristics, with more socially disorganized neighborhoods featuring more crime. Social disorganization, on the other hand, is be caused by factors such as poverty, high residential turn-over and ethnic heterogeneity, all of which hinder the formation of social bonds within the neighborhood, leading to said disorganization and then crime. It is worth noting, then, a conflict between social disorganization theory and others such as rational choice theory regarding the role of income. While the first predicts that crime is concentrated in poor areas, the second predicts the opposite. This conflict can be solved, at least in part, by noting that different types of crime feature not only different spatial patterns but may also have different relationships to income and related aspects, some being related to wealth and others to poverty. Moreover, the type of

connection may differ: Poverty (of the potential offender) is in many theories regarded as a motivator or strain factor, while wealth (of the potential target) is regarded as an attractor. This is related also to the problem of confounding areas of high crime occurrence with areas with high concentration of offenders living there, two sets that may not coincide. Finally, how far from one's own neighborhood an offender is likely to go to commit a crime may vary.

While these theories tend to approach crime in a more general sense, there are few a priori reasons to expect that a burglary would not fit the outlined theoretical frameworks. The economic driver related to income inequality theories of crime appears particularly fit for property crimes such as burglary, in which one of the practical outcome is a forced "transfer of income." Other crime theories described (e.g., routine activities, rational choice) show no obvious incompatibility with burglary, with this type of crime indeed being explicitly mentioned in some of these studies [7–10,27]. The question, however, is whether these theories that suggest a plausible link between localized income inequality and burglary are indeed reflected in the empirical geography of burglaries within cities.

Few studies found by me have so far investigated the relationship between income inequality and the spatial distribution of crime within cities. Most empirical studies that have tested the relationship between crime and income inequality have done so using macro-units of analysis such as whole cities, counties, regions or countries. Not only have these studies often disagreed regarding the nature of this relationship, few have investigated this potential connection at the within-city scale—that is, whether income inequality can explain the specific spatial patterns of crime observed within a city. Only five studies could be found that have done such an analysis [20–24]. Of these, only one analyzed the effect on property crimes, with the others focusing solely on violent crimes. Different types of crime often display distinct dynamics, that particularly being the case when comparing property crimes such as burglary with violent crimes like homicide. As such, only the study of Hipp, 2007 [21] can be considered to provide empirical evidence of the relationship between income inequality and the geography of burglaries within a city. In that study, the relationship between property crime (burglary and auto-theft) and income inequality was weak (albeit positive), contrasting with the stronger positive association detected between income inequality and violent crime. This again highlights the importance of more empirical studies to test potential relationships between income inequality and the geography of crime within cities.

To summarize, this subsection shows that a unified, explicit, and straightforward model relating income inequality to the spatial distribution of crime at a within-city scale is still lacking, but that multiple theories have offered elements that allow us to infer a potential relationship. In general, income has been theorized to have a dual effect: First, as a motivation (or strain) factor that presses the poorer to more likely consider crime as a money earning option; second, as an attractor, determining where opportunities for property crime are concentrated. Inequality should thus increase both of these aspects. Finally, the interaction between these two elements should be modulated by space, with interaction being more likely at smaller distances. Moreover, the perception of unattainable economic success described by strain theory can be theorized to be greater in a situation where the poorer and the richer are spatially closer. Therefore, according to this implied model, wealthier places are at greater risk than poorer ones, but, given two different wealthy locations, one closer to a poorer neighborhood is at greater risk. This theory has seldom been tested and fully formalized, in particular for within-city scales, and it is the objective of this study to contribute towards this goal. A proposed model is formally described in Section 2, with the results of it being applied to residential burglaries in Belo Horizonte being shown in Section 3. Section 4 concludes with a summary of these results and a comparison of the methodology employed (and associated results) to previous studies.

## 2. Materials and Methods

*2.1. Model Description*

The first question of this study concerns how much local burglary risk depends on local income levels. If we assume this hypothetical relationship to be linear, in quantitative terms it can be expressed as:

$$R_i = a_i I_i + o_i \tag{1}$$

$R_i$ Rate of burglary per residence at location $i$
$I_i$ Average household income at location $i$
$a_i$ Elasticity of $R_i$ in relation to $I_i$ and at location $i$
$o_i$ Component of $R_i$ that is independent of $I_i$

The second question concerns whether this relationship between burglary risk and higher income is boosted by the presence of nearby poverty. In more formal terms, the question is whether the coefficient linking burglary risk to income is itself dependent on local exposure to poverty (for simplicity, a linear relationship is assumed):

$$a_i = b_i E_i + c_i \tag{2}$$

$E_i$ Exposure to poverty at location $i$
$b_i$ Elasticity of $a_i$ in relation to $E_i$ and at location $i$
$c_i$ Component of $a_i$ that is independent of $E_i$

By combining Equations (1) and (2), we have:

$$R_i = (b_i E_i + c_i)I_i + o_i = b_i E_i I_i + c_i I_i + o_i \tag{3}$$

Finally, the third question concerns the scale at which exposure is most relevant. There are multiple ways that exposure can be defined and measured in the literature. In this study, the main interest is in "exposure to poverty" as a metric representing "how many poor residences are nearby," with the scale of exposure then being the question of what is considered by "nearby." With that general idea as a goal, exposure to poverty at location $i$ is then calculated as:

$$E_i = \sum_{j=1}^{n} W_{ij} P_j \tag{4}$$

$W_{ij} = 1 \; if \; dist(i,j) \leq \lambda$
$W_{ij} = 0 \; if \; dist(i,j) > \lambda$
$P_j$ Number of poor residences at location $j$
$\lambda$ Pre-defined threshold distance

The notion of nearby is therefore defined in a binary way, with all poor residences located within a bandwidth. By combining all the equations, we then have:

$$R_i = b_i I_i \sum_{j=1}^{n} W_{ij} P_j + c_i I_i + o_i \tag{5}$$

$W_{ij} = 1 \; if \; dist(i,j) \leq \lambda$
$W_{ij} = 0 \; if \; dist(i,j) > \lambda$

Finally, if we assume that the role of income inequality is the same across every location $i$ being considered, meaning that $b_i$ and $c_i$ are constant over space and can simply be denoted as $b$ and $c$, the final model is:

$$R_i = bI_i \sum\nolimits_{j=1}^{n} W_{ij}P_j + cI_i + o_i \tag{6}$$

$W_{ij} = 1 \; if \; dist(i, j) \leq \lambda$
$W_{ij} = 0 \; if \; dist(i, j) > \lambda$

Within this model, the answer to the research questions are contained in the values of $b$, $c$ and $\lambda$, as summarized below:

$b$ How much average income affects residential burglaries?
$c$ How much exposure to poverty affects residential burglaries, controlling for local income?
$\lambda$ What is the range at which exposure becomes significant?

The following section explains how these values can be estimated.

*2.2. Calibrating the Model*

If $\lambda$ is known, and we assume $o_i$ to follow a normal distribution, then $b$ and $c$ from Equation (6) can be estimated via linear regression from $R_i$, $I_i$ and $P_i$:

$$R \sim lm(IE, I) \tag{7}$$

$R = \{R_1, R_2, \ldots R_n\}$
$I = \{I_1, I_2, \ldots I_n\}$
$E = \{E_1, E_2, \ldots E_n\}$
$E_i = \sum\limits_{j=1}^{n} W_{ij}P_j$
$W_{ij} = 1 \; if \; dist(i, j) \leq \lambda$
$W_{ij} = 0 \; if \; dist(i, j) > \lambda$

Since the value of lambda is not known a priori, Algorithm 1 below can be used to estimate $b$, $c$ and $\lambda$. In short, the algorithm estimates $b$ and $c$ for a set of candidate $\lambda$ values, choosing the final set of $b$, $c$ and $\lambda$ from the model that provides the best fit (in a least-squares sense).

---

**Algorithm 1.** Algorithm to evaluate different bandwidths for exposure.

---

```
algorithm estimate_bclambda(lambda_candidates)
    R2_candidates = []
    b_candidates = []
    c_candidates = []
    for_each lambda in lambda_candidates do
        E = calc_E(P,lambda)
        regression_model = lm(R~IE,I)
        append(R2_candidates,regression_model.R2)
        append(b_candidates,regression_model.b)
        append(c_candidates,regression_model.c)
    end_for
    best_model = index_of_max(R2_candidates)
    b_final = b_candidates[best_model]
    c_final = c_candidates[best_model]
    lambda_final = lambda_candidates[best_model]
    return [b_final, c_final, lambda_final]
end_algorithm
```

---

*2.3. Control Variables*

Under the model just described, any effects not related to income are modeled in the factor $o_i$, which is assumed to be normal in distribution for the purposes of calibration. Additionally, linear regression assumes that the $o_i$ factor (treated as residual) is independent from the other explanatory variables (that is, $I$ and $IE$ in Equation (7)), which might not be the case. This assumption not being valid impacts the estimated values for the regression coefficients $b$ and $c$, as well as $\lambda$ indirectly.

To tackle this issue, control variables can be included in the regression to account for the portions of the variance not explained by the variables of interest in the study (see Appendix A for how the presence of spatial autocorrelation was tackled in this study). As such, the regression step then becomes:

$$R \sim lm(IE, I, C_1, C_2, \dots, C_k)$$

where $C_i$ is the i-th control variable included, with i = 1 ... k

The control variables that were included are listed below (Table 1), including a brief description of why the variable was included.

**Table 1.** List of control variables used, and their motivation.

| Variable | Motivation |
|---|---|
| Proportion of residences being rented | Proportion of rented residences is used here as a proxy for residential turn-over. Under the framework of social disorganization theory, higher residential turn-over is considered one of the factors leading to higher community disorganization and, ultimately, higher crime rates. |
| Proportion of land used for commercial establishments | Commercial activity is hypothesized to be relevant under two different theoretical frameworks. Under social disorganization theory, a higher mixing of commercial with residential may lead to higher community disorganization and, therefore, higher crime rates (This seems to be the most common understanding ([30], p. 4), although some studies ([31] cited in [30]) maintain that the greater presence of commercial establishments increases social cohesion, and thus, decreases crime). Under routine activity theory, the higher presence of commercial activity affects the flow of people in the region, possibly increasing the presence of potential offenders or potential informal guardians. |
| Number of bus stops | Number of bus stops is hypothesized to be relevant following the framework of routine activities theory, in which the presence of bus stops affects (or is related to) the flow of people in the region, possibly increasing the presence of potential offenders or potential informal guardians. Within the framework of crime pattern theory, the distribution of bus stops could work as a proxy for the aggregated activity spaces of individuals in a city. |
| Betweenness centrality | Similar to the number of bus stops, betweenness centrality could work as a proxy for modelling flow of people and their activity spaces, following the frameworks of routine activities theory and crime pattern theory. |
| Land-use density (real-estate units per $km^2$) | Similar to the number of bus stops and betweenness centrality, land-use density is a potential proxy for modelling flow of people and their activity spaces, following the frameworks of routine activities theory and crime pattern theory. |
| Distance to nearest police station | Distance to the nearest police station is used as a proxy for police deterrence. Following routine activities theory, for a crime to occur, a motivated offender and a suitable target must coincide in space and time with the absence of a capable guardian (i.e., the police). Greater distance to a police station is assumed here to be a potential indicator for reduced police presence. |

It is worth mentioning that the focus of this study was not to specifically investigate these control variables, and they were mainly included to reduce the possibility that any observed effects of income and exposure to poverty on crime would be proxies for these other variables.

*2.4. Data Sources*

2.4.1. Burglary

Burglary data used in this study originated from boletins de occorência, reports made to the police (Policia Militar de Minas Gerais) in the year of 2010 for the city of Belo Horizonte. A total of 6923 residential burglaries were reported to the police in this period in Belo Horizonte. These reports were geocoded into latitude and longitude, with an accuracy rate of 95% following the methodology proposed by [32].

2.4.2. Income data

Income data used in this study came from the Census of 2010 in Brazil, the latest census available up to this date. The finest unit in which data are organized and available is called the census sector (setores censitários), totaling 3936 census sectors in Belo Horizonte, with a median of 204 households, 552 residents and 49,619.18 m$^2$ in area. In this study, kriging interpolation was used to resample these census data into a uniform grid (see Section 2.4.4). Income information was provided in different formats in the census, two of which were used in this study. First, the total income for the whole census sector; second, the number of households per income per capita category within that census sector. The ten income categories used by the census are listed below (Table 2):

**Table 2.** List of income categories from the census of Brazil (Censo 2010).

| Category | Description (income per capita) |
|:---:|:---:|
| 1 | Less than 1/8 of a minimum wage |
| 2 | Between 1/8 and 1/4 of a minimum wage |
| 3 | Between 1/4 and 1/2 of a minimum wage |
| 4 | Between 1/2 a minimum wage and 1 minimum wage |
| 5 | Between 1 and 2 times a minimum wage |
| 6 | Between 2 and 3 times a minimum wage |
| 7 | Between 3 and 5 times a minimum wage |
| 8 | Between 5 and 10 times a minimum wage |
| 9 | More than 10 minimum wages |
| 10 | No income |

Since the total number of households per census sector was also given, total income per census tract could then be used to calculate the average household income of the census sector, which was then used by the model. Figure 1 maps the distribution of income in Belo Horizonte. The number of poor residences, also used in the model, was calculated by counting the number of residences in income categories 1–4, which approximated the range for the income classes D and E (the two poorer than middle-class), as defined by the Brazilian Institute of Geography and Statistics—IBGE [33]. In 2010, the minimum wage in Brazil was R$510 per month, approximately equivalent to U$300, using 2010's average exchange rate.

## Average household income per capita (in R$)

**Figure 1.** Average household income per capita (in Brazilian reais).

### 2.4.3. Data Sources for Control Variables

Table 3 lists the sources and types for the data used as control variables (areal, point or other). Figure 2 provides maps each of the control variables.

**Table 3.** Source and data type for the control variables used.

| Variable | Source | Type |
|---|---|---|
| Proportion of residences being rented | Population census of Brazil (Censo 2010) | Areal data (per census unit, then resampled to a uniform grid). |
| Proportion of land used for commercial establishments | Real-estate registry (IPTU) | Point data per commercial establishment (then aggregated to a uniform grid and proportions estimated). |
| Number of bus stops | OpenStreetMap | Point data (then aggregated to a uniform grid). |
| Betweenness-centrality | OpenStreetMap | Point and segment data (processed to generate values of betweenness-centrality and then aggregated to a uniform grid). |
| Land-use density (real-estate units per km$^2$) | Real-estate registry (IPTU) | Point data per real-estate unit (then aggregated to a uniform grid and density estimated). |
| Distance to nearest police station | Geocoded from addresses provided in the police's (PMMG) website | Point data for the police stations (the distance to nearest station was then calculated for each cell in a uniform grid). |

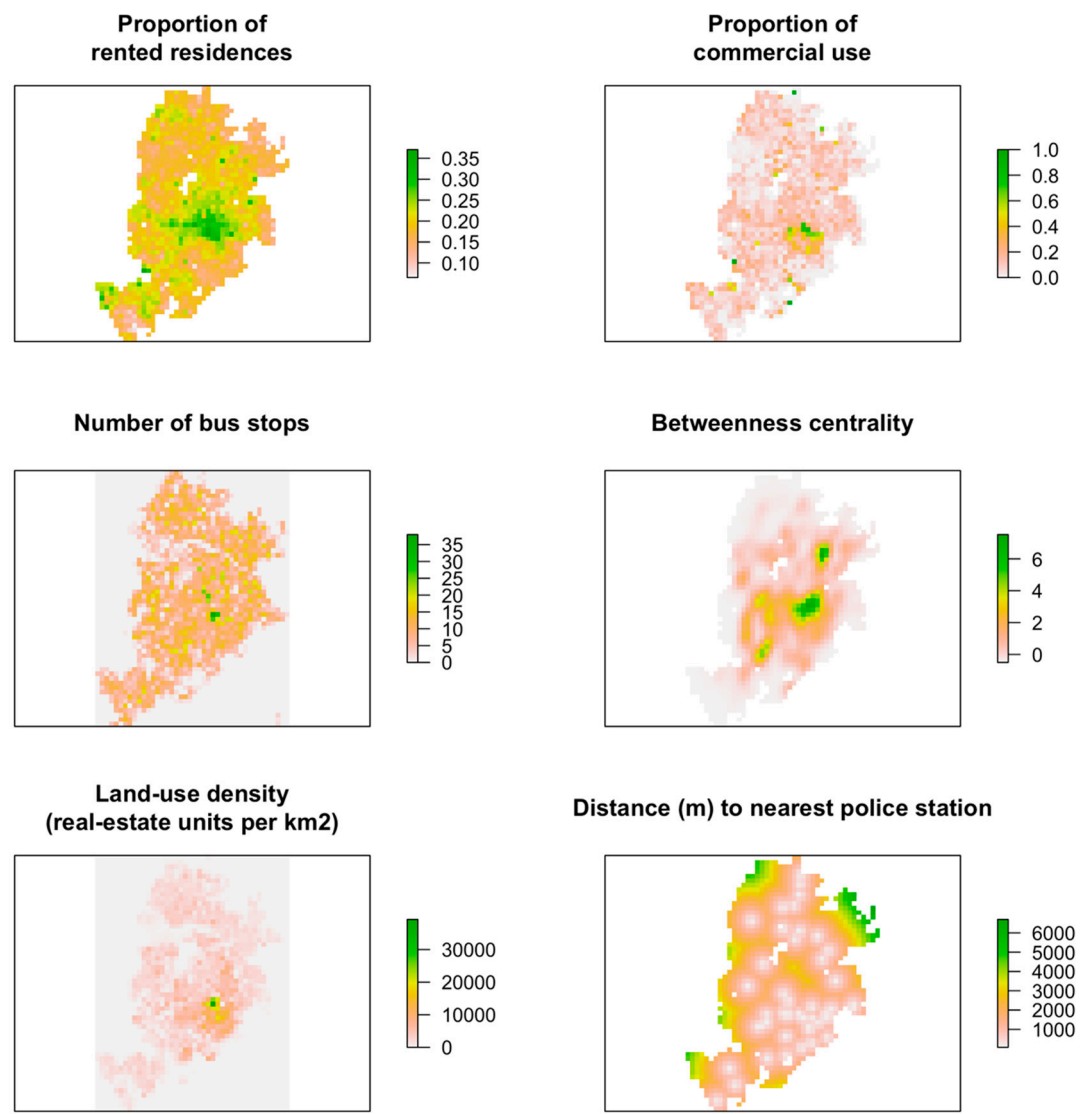

**Figure 2.** Mapping control variables.

### 2.4.4. Unit of Analysis Employed

A grid of 500 by 500 m square cells was used to map burglaries and other variables. A total of 1195 units were used out of a grid of 56 by 42 cells, of which some had to be discarded since they were outside the city limits (the considered study area). Burglaries and other point data were aggregated to appropriate cells, while areal data such as that from the census were interpolated using kriging. Figure 3 displays the map of burglary counts generated for Belo Horizonte.

Since this study was ultimately concerned with understanding crime, the observed spatial patterns of burglaries were used as a guide to decide on the unit of analysis for the study. The methodology described by Ramos et al., 2019 [34] was used to estimate a granularity that provided a balance between internal uniformity and robustness for the crime counts per areal unit, found to be 500 m. Internal uniformity is important to prevent problems due to the ecological fallacy: If crimes are not evenly distributed within an areal unit (which tends to happen when the unit used is too coarse), crime counts will be masking internal diversity and be a poor indicator of crime likelihood. If areal units are too fine, however, crime counts are less robust, and therefore a balance between the two criteria was sought.

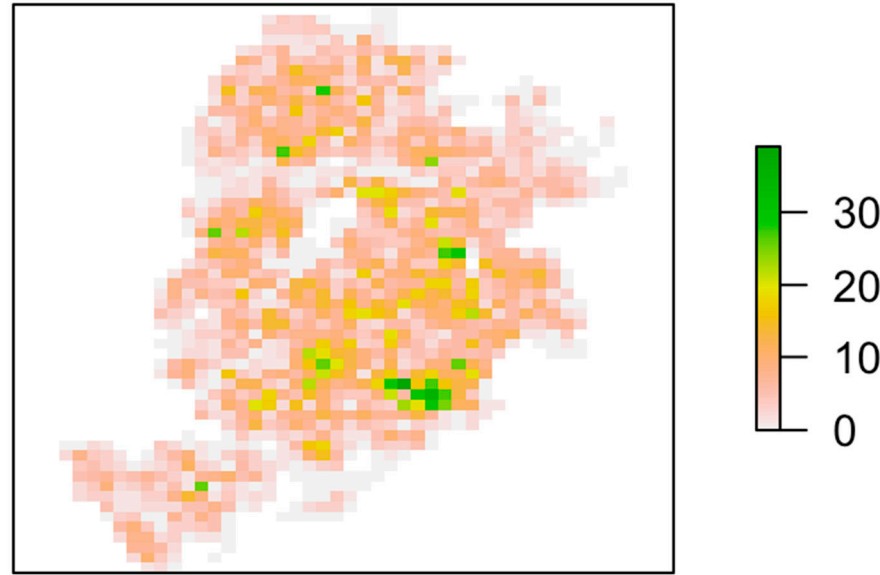

**Figure 3.** Burglary count for Belo Horizonte in 2010. Cell size used was 500 m.

2.4.5. Estimating Robust Rates of Burglary Per Residence

In this study, the GWRisk standardization method described by Ramos, 2019 [35] was employed to estimate the rates of burglary per house and per apartment from total burglary counts. The GWRisk method was shown to provide more reliable standardized estimates than other existing methods such as empirical Bayesian estimation or simple division of burglary counts by number of residence, as the GWRisk method is less prone to generating spurious peaks in areas with small populations. Moreover, contrary to these other methods, it allows for the estimation of separate standardized rates of burglaries per house and burglaries per apartment from plain burglary counts (not differentiated in terms of type of residence), as long as separate counts of apartments and houses are provided. Estimating separate risk maps was considered relevant in this study because it is possible that houses and apartments follow different dynamics concerning burglary (an assumption that was supported throughout this study). Figure 4 illustrates the maps of burglary risk obtained for Belo Horizonte, with Table 4 listing the mean and standard deviations values for both risks (for houses and for apartments), as well as other related variables. As seen in the table, the mean risk for houses was five times the mean risk for apartments.

**Table 4.** Mean and standard deviation values for burglary risk and related variables.

| Variable | Mean | Std. Dev. |
|---|---|---|
| Burglary Count (2010) | 5.587571 | 5.425531 |
| Number of houses | 166.4899 | 140.1903 |
| Number of apartments | 224.5036 | 414.9086 |
| Burglary risk for houses | 0.02434264 | 0.01090372 |
| Burglary risk for apartments | 0.004850637 | 0.008077198 |

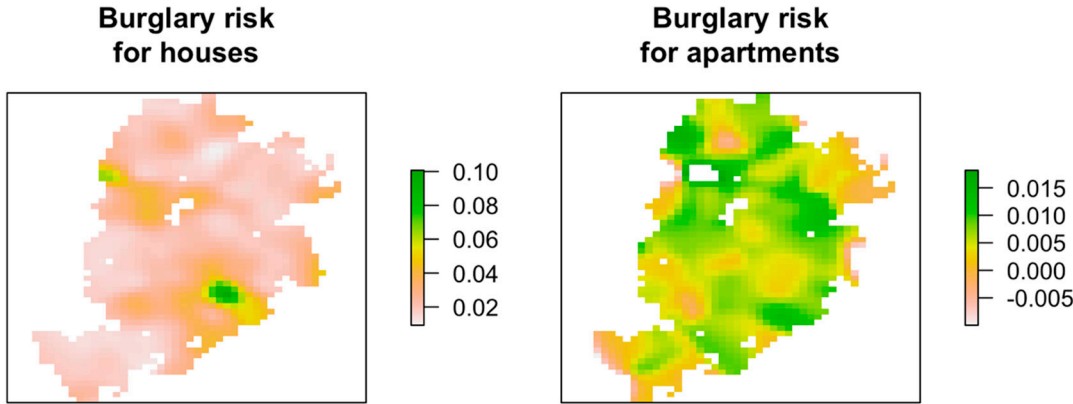

**Figure 4.** Maps of burglary risk for houses and apartments in Belo Horizonte (2010).

### 2.4.6. Study Area

Belo Horizonte is located in the state of Minas Gerais, which borders the states of São Paulo and Rio de Janeiro, where the cities of the same name are located (see Figure 5). Ranked as the third largest metropolitan area in Brazil (after São Paulo and Rio de Janeiro), the city proper of Belo Horizonte had in 2010 an estimated population of 2.375 million, with a demographic density of 7167 people per square kilometer in an area of 331 square kilometers. According to the 2010 Census (the latest available by this date), from a total of 628,447 households in Belo Horizonte, 66.58% are owner-housing units, 7.23% are in the process of being purchased, and 18.06% are rental housing units.

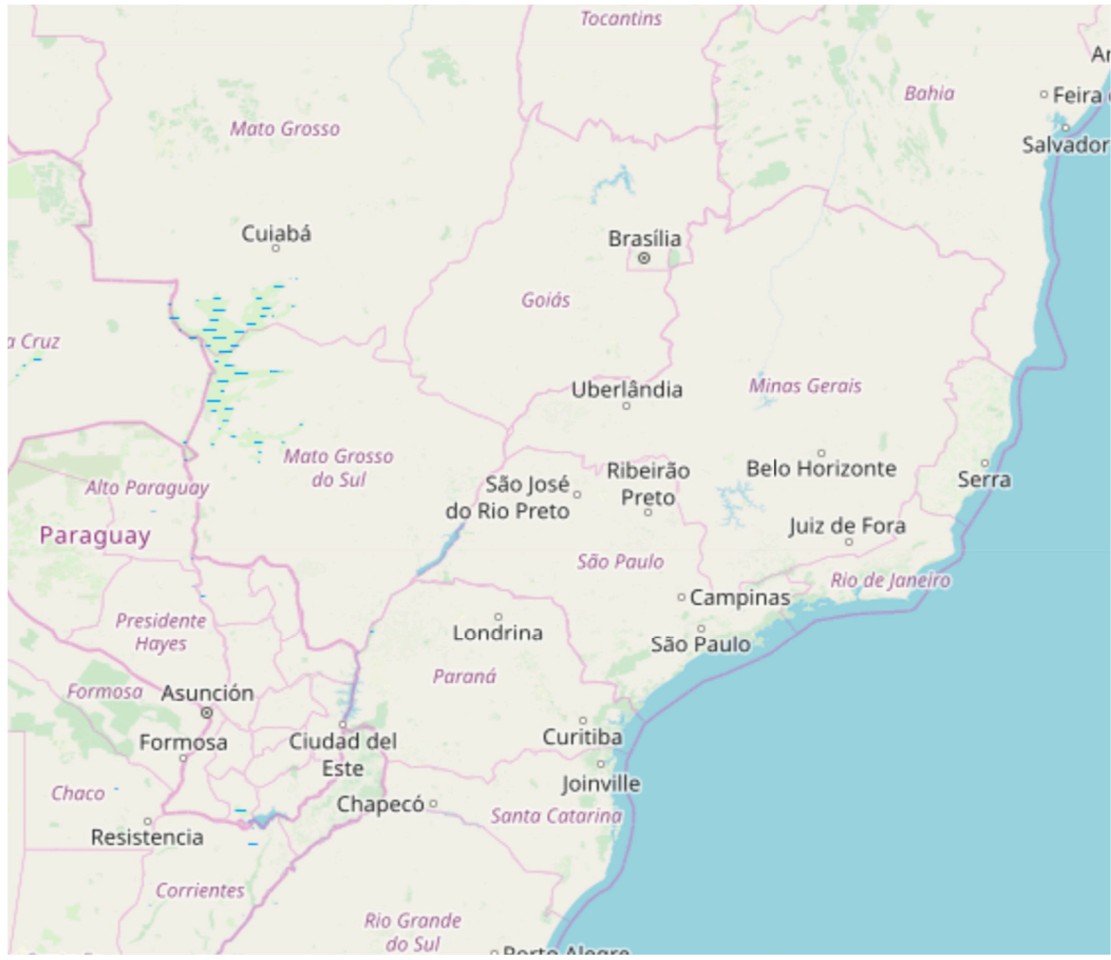

**Figure 5.** Map showing the location of the city of Belo Horizonte, Brazil. © OpenStreetMap contributors.

The population of Belo Horizonte typically lives in either residential apartments or houses. Houses are often detached or semi-attached, with an encircling wall or fence, including at the front of the house [36]. One exception to that trend are houses in gated neighborhoods (condomínios fechados); however, not only are these a small minority overall, they are also mostly located in the metropolitan area surrounding the city proper of Belo Horizonte [37], a region which was not considered in this study. Apartments are located in high rise buildings that are usually detached and gated, often having a doorman. Approximately 30% of the population of Belo Horizonte lives in these residential apartments [36]. Additionally, approximately 10% of the population in Belo Horizonte lives in favelas (slums). Residences in these informal settlements vary in quality, ranging from improvised shacks to brick-and-mortar constructions; in general, though, they can be classified as houses instead of apartments or some other type of residence. However, since these are often irregular residential areas, residences in favelas are often not listed in the real-estate registry used in this study. While this could be a source of bias for this study, ethnographic studies [38,39] have indicated that property crimes (e.g., burglary) are not particularly frequent in favelas, with most crimes being violent crimes (homicides) or drug-related crimes, an observation that was supported by the reported burglaries data used in this study. As such, it seems reasonable to consider this potential bias to be small.

## 3. Results

The estimated coefficients for average household income and exposure to poverty, varying with bandwidth size, as well as their corresponding *p*-values, are shown in Figure 6 for the case of single family houses. Figure 7 shows how much the model fitness varied according to the bandwidth used: The graph on the left-hand side shows the R-squared value for the regression using different bandwidths, while the graph on the right shows the normalized values of three different fitness scores—R-squared, F-score, and Akaike information criterion—varying with the bandwidth.

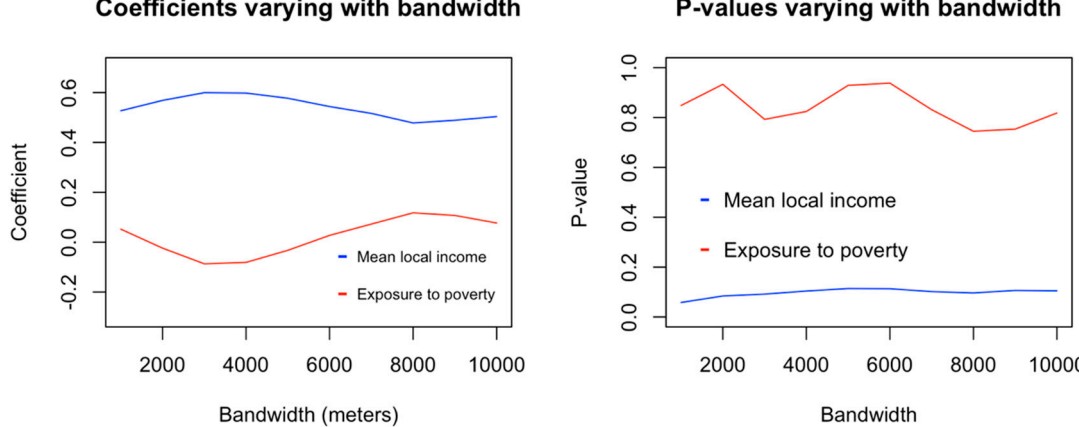

**Figure 6.** Coefficient values and their significance varying with bandwidth for the case of burglary risk at houses.

As can be seen from Figure 6, no statistically significant influence from exposure to poverty was found on burglary risk for houses at any of the bandwidths tested. Average household income, on the other hand, showed a high statistical significance, with higher income being associated with higher burglary risks. On average, a 10% increase in wealth is associated with a 5% increase in burglary risk for houses (using standardized units). As seen on Figure 7, the regressed model showed greater least-squares fitness at two different local peaks, one for a bandwidth of 1000 m and one for a bandwidth of 8000 m. Alternative fitness metrics used (F-score and Akaike information criterion) showed a similar pattern. Nevertheless, R-squared values were very similar across bandwidths, and since exposure to poverty was observed to be not significant at any bandwidth, these differences of fitness can also be considered insignificant.

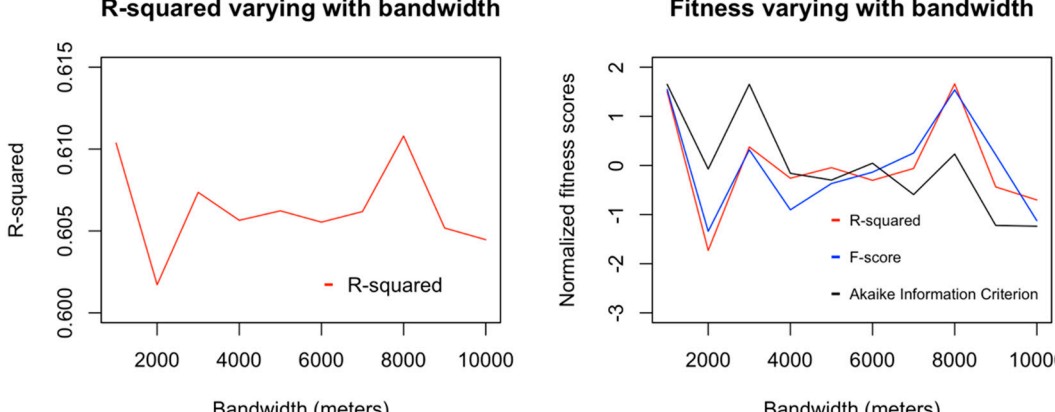

**Figure 7.** Model fitness varying with the bandwidth chosen for the case of burglary risk at houses. To the left, the R-squared metric is shown; to the right, three fitness scores (R-square, F-score, Akaike information criterion), normalized in value, are compared.

Finally, Table 5 shows the coefficient for all variables used estimated using a bandwidth equal to 8000 m, which yielded the highest fitness overall.

**Table 5.** Estimated model for burglary risk at single family houses. Regressed coefficients and corresponding *p*-values are shown for all variables considered, using a bandwidth $\lambda$ equal to 8000 m (which provided the highest fitness score).

| Variable | Standardized Coefficient | *p*-Value |
|---|---|---|
| Average household income | 0.4782948 | 0.09633432 * |
| Exposure to poverty | 0.1176405 | 0.7449263 |
| Proportion of residences being rented | −0.09458062 | 0.652599 |
| Proportion of land-use used for commercial establishments | 0.006974763 | 0.9811022 |
| Bus stops | 0.04087194 | 0.8617972 |
| Betweenness-centrality | −0.03537825 | 0.8603084 |
| Land-use density (real-estate units per km$^2$) | 0.000834463 | 0.9977056 |
| Distance to nearest police station | −0.01891306 | 0.8773111 |
| **R-squared** | 0.6107988 | |
| **Sample size** | 1195 (in average, a subsample of 100 units was used per iteration, as described in Algorithm A1) | |

The same study was then repeated for burglary risk for apartments. Figures 8 and 9 show the variation of fitness, coefficient value and *p*-values at the various tested bandwidths.

In the case of burglary risk for apartments, neither average household income nor exposure to poverty were observed to influence risk in a statistically significant sense. In general, fitness tended to increase with the bandwidth; however, since the statistical power remained too weak, this bandwidth effect can be also considered insignificant. The regression coefficients and their statistical significance are shown in Table 6.

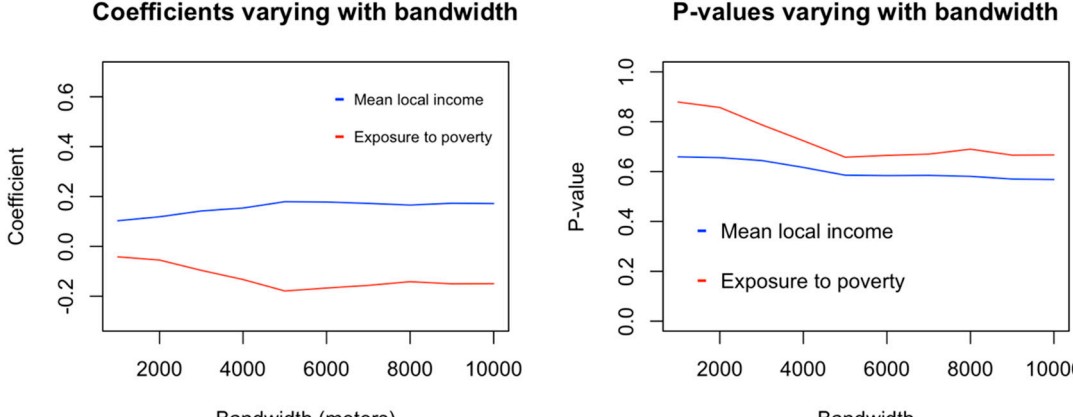

**Figure 8.** Coefficient values (**left**) and their significance (**right**) varying with bandwidth for the case of burglary risk at apartments.

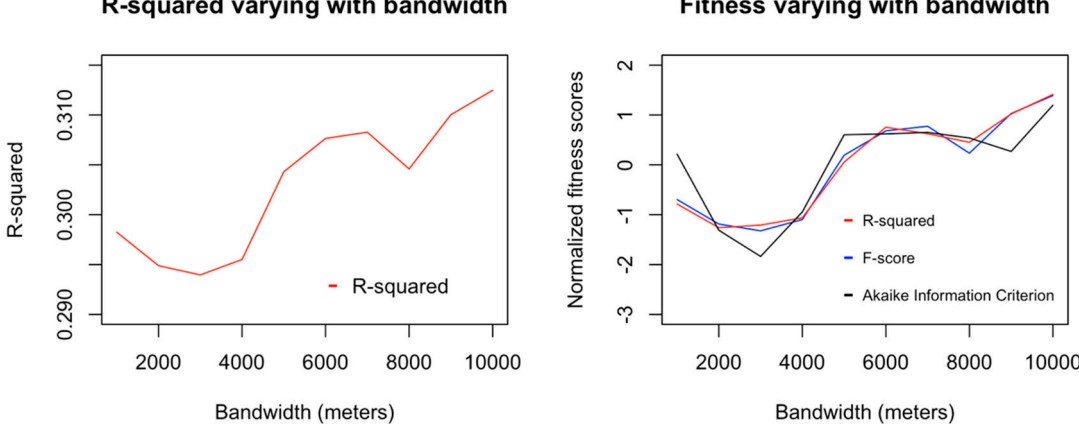

**Figure 9.** Model fitness varying with bandwidth chosen for the case of burglary risk at apartments. To the left, the R-squared metric is shown; to the right, three fitness scores (R-square, F-score, Akaike information criterion), normalized in value, are compared.

**Table 6.** Estimated model for burglary risk at residential apartments: Regressed coefficients and statistical significance of all variables used for a bandwidth of 9000 m (which provided the highest fitness score).

| Variable | Standardized Coefficient | *p*-Value |
|---|---|---|
| Average household income | 0.1717768 | 0.5682183 |
| Exposure to poverty | −0.1496601 | 0.6667372 |
| Proportion of residences being rented | 0.04697557 | 0.8319795 |
| Proportion of land-use used for commercial establishments | −0.03674683 | 0.8976192 |
| Bus stops | −0.02846617 | 0.9016353 |
| Betweenness-centrality | 0.07614556 | 0.6844067 |
| Land-use density (real-estate units per km$^2$) | 0.05594488 | 0.8198754 |
| Distance to nearest police station | −0.2371203 | 0.3029276 |
| **R-squared** | 0.3124782 | |
| **Sample size** | 1195 (in average, a subsample of 100 units was used per iteration, as described in Algorithm A1) | |

Finally, based on these results, the answers to the specific questions investigated by this research (for the case of residential burglaries in Belo Horizonte) are the following:

1: What is the effect of local average income level to local burglary risks?

Average household income is positively related with burglary risk for houses. On average, a 10% increase in household income is related to a 5% increase in burglary risk for houses, using standardized units. However, average household income shows no statistically significant relationship with burglary risk for apartments.

2: What is the effect of higher exposure to poverty on burglaries when local average income level is controlled?

A higher exposure to poverty has no statistically significant effect on burglary risk for either houses or apartments.

3: What is the scale at which this exposure is most relevant?

The scale of exposure is irrelevant, since exposure at any scale was not observed to be significant, both for houses or apartments.

## 4. Discussion

The tested model succeeded in explaining approximately half of the variance in the observed distribution of burglary risk for houses. The observed distribution of burglaries in Belo Horizonte appears to be significantly related to income, with the risk of burglaries for houses being higher in areas with an higher average household income. For the case of apartments, however, a link was not found, and the variance explained by the model was small. Nevertheless, although the positive link between burglary risk and income exists only for houses and not for apartments, burglaries of houses correspond to the majority of cases in Belo Horizonte, both in absolute counts and in a relative (per residence) sense, highlighting the importance of these findings.

No evidence was found that exposure between different income groups increases burglary risk (either for houses or apartments). If local income level was fixed, increasing the local exposure of that area to poverty showed no significant effect on burglary risk. This was true for different spatial scales of exposure. In other words, the burglary risk for a wealthy house is indifferent to whether there are poor neighborhoods nearby or not, at least in this study. While in a limited sense, income inequality could be said to be connected to crime in that risk is different for different income groups, that is not usually the sense implied when a connection between inequality and crime is referred to.

These results by themselves do not necessarily disprove the theories that connect crime to inequality in a general sense. However, they do show that there is no obvious observable connection between burglary risk and exposure among different income groups for the case of Belo Horizonte. Rather, if burglary tends to be more likely in higher income areas but the presence of nearby poorer neighborhoods is irrelevant, that could plausibly be explained in terms of the distribution of opportunities: More income meaning more objects of value to be stolen. This theoretical explanation would also be consistent with houses being more at risk than apartments, since in Belo Horizonte (as in other Brazilian cities in general), residential apartments are generally considered safer because they more often feature security mechanisms such as doormen and security cameras. Another plausible factor that may explain this observed irrelevance of localized income inequality on burglary risk would be burglars (hypothetically) having a high mobility, not being limited to nearby wealthy locations or even avoiding them to avoid suspicion. Additionally, a different possibility would be that burglars do not come from the poorest backgrounds but have a different socioeconomic profile, and so the proximity to very poor neighborhoods becomes insignificant to burglary risk. Additional research on the social profile and modus-operandi of burglars in Brazil could aid in clarifying these possibilities.

In addition, no significant relationship was detected between burglary risk and any of the control variables, either for houses or apartments. These results again give weight to the explanation that the potential gain for burglars is the prime factor governing the spatial distribution of burglary risk. Variables that are plausible proxies to pedestrian flow and urban accessibility, such as the number of bus-stops, betweenness-centrality, commercial presence, and land-use density yielded no significant correlation—a connection that would, however, be expected under the framework of routine activities theory. Similarly, the proportion of rented residences yielded no correlation to risk, and income yielded

a positive correlation to risk, both conflicting with what would be expected from an explanation based on social disorganization theory.

Whether these results can be generalized to other cities, however, would require repeating the study in different study areas. Compared to the results of Hipp, 2007 [21], these results differ significantly. It is worth noting, however, that the methodologies differ significantly as well: In the study by Hipp, 2007 [21], burglaries were not separated according to type of residence, the methods used for estimating burglary risk differed, and finer units of analysis were employed in this study. This is not a demerit of Hipp, 2007, since that study was not specifically tailored for analyzing burglary and income inequality, being focused on other questions (ethnic heterogeneity and multiple types of crimes), but it does justify why a different methodology was chosen in this study. Lastly, another important difference concerns the specific metric chosen for measuring income inequality. Contrasting to that of Hipp, 2007 and other studies [20–24] in which an income inequality metric such as the Gini index was used, this study modelled the effect of income inequality by having a (spatially lagged) exposure to poverty variable act as an interactive variable which boosted the effect of local household income (another independent variable in the model). This design was chosen in order to explicitly model the inequality effect described in most theories (e.g., economy of crime, relative deprivation and strain theories), with wealth acting as an attractor to crime and poverty acting (at distance) as a booster to such crime rates. Using an inequality metric such as the Gini index, on the other hand, can generate results that are harder to interpret within the context of this research. Due to income distribution being often more concentrated at the upper end, the income gap between the poor and the middle class is smaller (at least in numeric terms) than the gap between the middle class and the rich. As such, areas mixing middle-class and upper-class households tend to feature a higher inequality index (e.g., Gini index) than areas mixing middle-class and poor households. While this type of inequality could be hypothesized to be relevant, it is nevertheless different from the usual sense of inequality most often present in criminological theories in which poverty is a significant component. Having mean income as an independent variable can aid in controlling for this correlation between income inequality and wealth, but the difficulty remains of discerning whether inequality is more related to poverty or to wealth. The same reasoning (ease of interpretation and avoiding confounding results) applies to other model design choices such as the choice of not using segregation metrics like those proposed by Reardon and O'Sullivan, 2004 [40] or Feitosa et al., 2007 [41].

A potential limitation of this study is the presence of underreporting in crime rates and whether there are any biases associated with it—a common challenge with many crime studies. Burglary data used in this work came from the police records of reported cases; however, not all the cases are necessarily reported to the police, with the reporting rate for burglaries in Belo Horizonte having been estimated as 33% [36]. Since the main interest of this study was in the spatial distribution of crime and the factors explaining it, the main concern was not so much about the absolute numbers of crime rates but whether there are any biases on their spatial distribution and whether these biases are related to any of the explanatory factors considered in the analysis. For instance, could the positive relationship observed between burglary risk for houses and income be a product of wealthier citizens reporting more often? One possible way to determine the presence of such biases is conducting a victimization survey; unfortunately, however, I have found no such survey for the specific case of residential burglaries in Belo Horizonte (or in Brazil as a whole). A victimization survey [42] conducted for other types of crime in Brazil did provide some information, showing that there is both a higher reporting rate from wealthier citizens and a higher victimization rate for property crimes such as theft and robbery. As such, if burglaries in Belo Horizonte do follow the same pattern observed for other property crimes in Brazil as a whole, then this positive relationship between burglary risk and income could be a genuine one, albeit one boosted by the different reporting rates. However, a more specific victimization survey focused on burglaries in Belo Horizonte (or at least in Brazil) would be required to better gauge the potential biases from underreporting.

Finally, there is the question on whether the connection between income, poverty and inequality to crime as modelled and tested in this paper is justifiable. For instance, issues such as the penalization of poverty [43,44] raise a cautionary note not only on the potential analytical biases of linking poverty to crime but also on the more practical and ethical ramifications of considering such a hypothetical link. On the other hand, there is a range of theories that give some support to that assumption (as described in the introduction). This study attempts to be neutral on that respect, testing a hypothetical link that is recurrent outside of academia (justifiably or not) and that is supported or suggested by current criminological theory. The outcome of this study is that this hypothetical link was not observed, which, while not sufficient to disprove a link between crime and inequality in a more general and absolute sense, is an indicator that such relationship is more nuanced, perhaps less spatially explicit, and possibly more dependent on the type of crime considered.

In summary, this paper has proposed a quantitative model to evaluate the relationship between income inequality and the geography of crime within cities. The model implements a combination of mechanisms described or suggested by theories relating crime to income inequality, with local wealth acting as an attractor to crime and surrounding poverty acting as a boosting factor. The model was applied to residential burglaries in the city of Belo Horizonte, Brazil. The study has shown that mean local household income is significant and positively correlated to increased burglary risk for houses, explaining 61% of the variance; the presence of nearby poverty is irrelevant. These results and others suggest that the observed geography of residential burglaries is shaped by the distribution of opportunities and potential gains for burglars, and this geography is not a product of localized income disparity. This type of study has seldom been done at a within-city scale, in particular for property crimes.

**Author Contributions:** Conceptualization, Rafael Ramos; Methodology, Rafael Ramos; Software, Rafael Ramos; Validation, Rafael Ramos; Formal Analysis, Rafael Ramos; Investigation, Rafael Ramos; Resources, Rafael Ramos; Data Curation, Rafael Ramos; Writing-Original Draft Preparation, Rafael Ramos; Writing-Review & Editing, Rafael Ramos; Visualization, Rafael Ramos; Supervision, Rafael Ramos; Project Administration, Rafael Ramos; Funding Acquisition, Rafael Ramos.

**Funding:** This research is part of a PhD funded by the Science Without Borders fellowship from CAPES (Coordenação de Aperfeiçoamento de Pessoal de Nível Superior).

**Acknowledgments:** The author would like to thank the anonymous reviewers, as well as Keith Clarke, Helen Couclelis, Bráulio Silva and Flávia Feitosa for valuable comments and suggestions for this paper. The author would like to thank Keith Clarke for proving funding for the publication of this paper. The author would also like to thank Haiyun Ye, Marcela Suárez, Mike Johnson and S. Lucille Blakeley for proof reading part of this paper and offering valuable suggestions. Finally, the author would like to thank CAPES (Coordenação de Aperfeiçoamento de Pessoal de Nível Superior) for funding the PhD of which this research is part of.

**Conflicts of Interest:** The author declares no conflict of interest.

## Appendix A

*Controlling for Spatial Autocorrelation*

The issue of controlling for spatial autocorrelation is a problem present in any regression analysis with spatial data. In the current literature, there has been a diversity array of approaches for dealing with spatial autocorrelation in regression analysis [45,46]. One approach is to model spatial autocorrelation in the regression model, leading to spatial autoregressive models such as the spatial lag and spatial error models. One difficulty with this type of approach is that it involves a choice for a specific way of modeling this spatial effect in order to control for it; this choice, on the other hand, relies on making an assumption about the phenomena being investigated, and there is the possibility that an incorrect model is chosen [47,48].

The approach chosen in this study was to subsample the original dataset so that the sample units were distant enough to be free of spatial autocorrelation before conducting the regression analysis. The distance at which spatial autocorrelation was considered to be absent was estimated from the range of the variogram generated from the original dataset for burglary risk. Since multiple different

subsamples are possible, the regression was conducted for a large number of different subsamples, with an average set of regression coefficients (and their significance) being estimated in the end. The following Algorithm A1 details this process:

---

**Algorithm A1.** Algorithm subsamples the original dataset so that units are distant enough and not spatially autocorrelated. Regression was performed with multiple different subsamples to estimate an average final result.

---

```
algorithm subsampled_regression(regression_data,iterations)
    variog = variogram(regression_data.y)
    b_sub = vector(length=iterations)
    c_sub = vector(length=iterations)
    R2_sub = vector(length=iterations)
    for i in 1 to iterations do
        sub = subsample(regression_data,dist >= variog.range)
        sub_regression_model = lm(sub)
        b_sub[i] = sub_regression_model.b
        c_sub[i] = sub_regression_model.c
        R2_sub[i] = sub_regression_model.R2
    end_for
    b = mean(b_sub)
    c = mean(c_sub)
    R2 = mean(R2_sub)
    return [b,c,R2]
end_algorithm
```

---

There is a tradeoff for this method in that using a smaller sample size may lead to less accurate results. However, since the subsample in this case was still large enough (approximately 100 units) for doing reliable statistical inferences, this tradeoff can be considered mild compared to how a bias of choosing an incorrect spatial model could affect the estimated coefficients of the explanatory variables.

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
