# Peer review of "Does Income Inequality Explain the Geography of Residential Burglaries? The Case of Belo Horizonte, Brazil"

_ijgi, doi:10.3390/ijgi8100439_

Round 1
Reviewer 1 Report
I urge the author to shorten and sharpen the paper.
Author Response
Point 1: I urge the author to shorten and sharpen the paper.
Response 1: Thank you for your feedback. I have moved the subsection about controlling for spatial auto correlation to an appendix. I believe this is a methodological detail that should be kept in the manuscript but that could be moved out of the main text to make it shorter and improve the flow. Since other reviewers have requested me to add more details in some other aspects of the study, I could not reduce its length any further.
Thank you again for your comments, I am open to provide further improvements if necessary.
Reviewer 2 Report
This manuscript explores the spatial relationships between burglaries and income inequality within the environment of urbanized areas. The model produced medium-strength significant relationships between burglaries per houses and lower-income. The manuscript could use some improvement in the method section.
The author asked three questions. The first two questions asked how independently income level and poverty are related to burglaries. However, in following the formula (line 185), the income level became dependent on the poverty level, which was disconnected from the research question. Furthermore, the spatial unit of the collected income data was different from that used in the analysis. How did the author match the two datasets when the spatial units are different? On line 344, the author classified the income data into ten categories, while only average income was used in the later results. Also, why differentiated burglaries per houses and that per apartments? Was it only single-family houses were used? The author needs to provide more description of the study area.
In addition, the manuscript needs other minor revisions in the following areas:
The theories cited in the article discussed crime in a general sense. More in-depth discussion (and literature review) is needed about the specific relationships that poverty/burglaries crime has with income in order to make the article’s contribution clearer. Give a brief introduction of the study area to help the readers understand its urban structure.
Overall, the article fits within the journal while needs further improvement.
Author Response
Point 0: This manuscript explores the spatial relationships between burglaries and income inequality within the environment of urbanized areas. The model produced medium-strength significant relationships between burglaries per houses and lower-income. The manuscript could use some improvement in the method section.
Response 0: Thank you for your feedback, I believe you raise some fair points. I have updated the manuscript following your (and the other reviewers’) comments. See below my responses to the specific points mentioned.
Point 1: The author asked three questions. The first two questions asked how independently income level and poverty are related to burglaries. However, in following the formula (line 185), the income level became dependent on the poverty level, which was disconnected from the research question.
Response 1: I believe the wording (in particular the use of the term “independently”) might have lead to some ambiguity. I have rewritten them to be:
What is the effect of local average income level on local burglary risk? What is the effect of higher exposure to poverty on burglaries, while controlling for local average income level? What is the scale at which this exposure is most relevant?Another way (less general) of wording the research questions would be: do higher income locations feature higher burglary risk? Given a higher income location, will the presence of nearby surrounding poverty increase the risk of burglary at that higher income location? What is meant by nearby, and at what scale is the effect of proximity to poverty relevant, if any? In this case then, exposure to poverty and local income are conceptually different and independent, since local income is based on the income level of location i only, but exposure to poverty is based on the poverty surround location i, so that in formal terms, there is no dependence in the equation but an interaction between local income level and exposure to poverty. There is some degree of statistical correlation due to the way income tends to be distributed and clustered, but they are different concepts. I have added to the manuscript a version of this “in other words” paragraph right after the research questions to exemplify and clarify these points. Please let me know if that did not address your question/comment.
Point 2: Furthermore, the spatial unit of the collected income data was different from that used in the analysis. How did the author match the two datasets when the spatial units are different?
Response 2: As mentioned in subsection 2.4.4 (Unit of analysis employed), the census data (originally polygons) was interpolated using kriging to a uniform grid of square cells. The cell size itself was determined based on the properties of the burglary point pattern, using the methodology from Ramos et al (2019). Using this methodology, burglaries are mostly uniformity distributed inside each cell. I have now added a mention to that also in subsection 2.4.2, and also specific that kriging was used for interpolation.
Point 3: On line 344, the author classified the income data into ten categories, while only average income was used in the later results.
Response 3: Income information from the census was available in two forms: total income of the census tract and number of households in each of the ten income categories. Notice that these ten income categories were not defined my me (the author) but by the census itself. I did some edits in section 2.4.2 to make this clearer.
Additional details for why I chose to use the two different formats is given below:
Since the first research question is concerned with average income level, I calculated it by diving the total income of the census unit by the number of household. This seemed to be the most straightforward approach, although I did consider some alternative design. For instance, I considered using the proportions within each of the ten income groups to represent the local income profile, however this would increase too much the number of variables, which would also be correlated to each other to a significant degree, leading to ambiguous results. Also, selecting only a few of these income categories to represent average income level would require some subjectivity in the selection. Finally, I did consider some form of principal component analysis to summarize the income category variables into one, however, using average income just seemed more straightforward.
For the second research question, the focus is on the presence of nearby poverty, therefore using the number of household per income category seemed more appropriate, since it is not possible to rule out that an average income value indicating middle-class would contain also poor (and/or rich) household.
Finally, I’d like to mention that I did test these alternative different model designs, and the results were typically not too different. However, the interpretation of the results was clearer and less ambiguous in the current design, which is why it was preferred.
Similar considerations were already included in the Discussion section, and since one of the reviewers has requested the paper to be shortened, I decided to not include further details for the sake of brevity.
Point 4: Also, why differentiated burglaries per houses and that per apartments? Was it only single-family houses were used?
Response 4: Yes, the term houses is used meaning “single family houses”. I have added a clarification to that in the introduction now. Separate risks for houses and apartments were considered because it was hypothesized that their risk maps (and potential explanations for it) would be different (a hypothesis that the results seem to support). I have also included a mention to that in the methodology
Point 5: The author needs to provide more description of the study area.
Response 5: I have now included more details on the urban area, as well as a map locating the city of Belo Horizonte, Brazil.
Point 6: In addition, the manuscript needs other minor revisions in the following areas:
The theories cited in the article discussed crime in a general sense. More in-depth discussion (and literature review) is needed about the specific relationships that poverty/burglaries crime has with income in order to make the article’s contribution clearer.
Response 6: I have added a paragraph in the Literature review discussing how the theories described apply (in a more logical/theoretical sense) to burglaries, and that the question is whether this plausible link between localized income inequality and burglaries is indeed reflected in the empirical observed geography of burglary within cities (leading then to the discussion about the few empirical studies conducted on this topic).
Point 7: Give a brief introduction of the study area to help the readers understand its urban structure.
Response 7: Additional details on the study area have now been included.
Thank you again for your valuable comments. I am open to provide further improvements if necessary.
Reviewer 3 Report
Dear Author, thank you for your paper, it was a delight to read it. What a paper after so many boring ones.
I do not claim to understand your model in full depth, but with my knowledge and skills, it seems fine.
I may have some minor comments regarding the text.
It looks a bit weird using "we" in the abstract, while you are the only author of the paper. I would also take "income inequality" out of the keywords, as it is already mentioned in the title of the paper.
In the Introduction, line 41, you speak about "only three studies" - well, I did not do any intensive search, but even if you find only 3 research papers, it does not mean, that there are only three studies - maybe only three studies, published in English, indexed on WoS - yes, but surely there may be some unpublished studies, in Chinese, whatsoever. I hope you understand my point, it is not about patronising you, just a reminder to be cautious about wording.
It would be great if you can also provide a map locating your case study location.
I was also wondering, why did you use data from 2010 - do I get it right, it is the last census data?
Why are maps in Fig 2 having a different shape? Is it a scale? projection?
In the discussion, you talk about wealthier houses being robed more often, but you do not reflect anywhere in the paper, that you are working with a registered crime only - meaning, that poorer households may not even border to report the crime? Would not that be a limitation of your research?
Somehow, the footnotes are all numbered as "1", maybe it is just an issue with my version, but it is a bit confusing.
As I said in the beginning, I really liked your paper and I am sure these minor comments are easy to deal with.
Author Response
Introduction: Dear Author, thank you for your paper, it was a delight to read it. What a paper after so many boring ones. I do not claim to understand your model in full depth, but with my knowledge and skills, it seems fine. I may have some minor comments regarding the text.
Introduction (Response): Dear Reviewer, thank you very much for your comments and kind words. I have updated the manuscript to conform to your comments and suggestions, but I’m willing to make other changes if I missed any aspect. See below for my specific responses and updates.
Point 1: It looks a bit weird using "we" in the abstract, while you are the only author of the paper. I would also take "income inequality" out of the keywords, as it is already mentioned in the title of the paper.
Response 1: I have now replaced “we” for the passive voice, except in parts that involve mathematical formulations (such as “if we consider…”)
Point 2: In the Introduction, line 41, you speak about "only three studies" - well, I did not do any intensive search, but even if you find only 3 research papers, it does not mean, that there are only three studies - maybe only three studies, published in English, indexed on WoS - yes, but surely there may be some unpublished studies, in Chinese, whatsoever. I hope you understand my point, it is not about patronising you, just a reminder to be cautious about wording.
Response 2: That is a fair point. I have reworded that part to be more cautious about it. I have also corrected it to be five papers (since in the literature review I mention a total of five papers)
Point 3: It would be great if you can also provide a map locating your case study location.
Response 3: I have now added a map showing the location of the study area.
Point 4: I was also wondering, why did you use data from 2010 - do I get it right, it is the last census data?
Response 4: Yes, that is the latest census. I have now included a mention of that in the manuscript.
Point 5: Why are maps in Fig 2 having a different shape? Is it a scale? projection?
Response 5: They got slightly different shapes due to how R plotted the maps and handled aspect ratio. All maps have the same projection and use the same grid extent and number of cells. I have now been able to correct this slight change in size.
Point 6: In the discussion, you talk about wealthier houses being robed more often, but you do not reflect anywhere in the paper, that you are working with a registered crime only - meaning, that poorer households may not even border to report the crime? Would not that be a limitation of your research?
Response 6: Yes, that is a possibility and again a fair point. To my knowledge there is only one victimizations survey done in Brazil that provide some insight into that (though not perfect). In general, property crimes tend both to victimize the wealthier and be more reported by them. This survey did not specifically approach burglaries, asking instead about other property crimes like theft and robbery. However, if burglary is to follow the same pattern, then this positive relation between burglary risk for houses and income is likely to be a real one, albeit boosted by the biased underreporting. I have now added a paragraph commenting on that in the Discussion section.
Point 7: Somehow, the footnotes are all numbered as "1", maybe it is just an issue with my version, but it is a bit confusing.
Response 7: I had set the numbering to restart at every page (I think my understanding was that this was the appropriate way!). I have changed the numbering to be continuous not.
Conclusion: As I said in the beginning, I really liked your paper and I am sure these minor comments are easy to deal with.
Conclusion (response): Thank you again for your valuable comments! I am open to provide further improvements if necessary.
Reviewer 4 Report
The article presents a consistent and interesting research, with well defined aim and study area, and good theoretical background. I have only more general comment concerning the study’s assumptions. In the article, there is a suggestion that poverty is a cause of crime, which is supported by some theories. However, there is also a scientific discussion on stereotypes concerning enclaves of poverty, here mentioning only Wacquant’s poverty penalization, stigmatization of urban subploretariat and connected to them social and security policies. Deprived urban spaces treated as spaces of danger and violence with the poor treated as potential criminals is a rather simplistic approach. I understand the study presents only an excerpt of a research on social inequalities and on a geography of crime but in my opinion a comment is needed in theoretical background and in discussion sections on the complexity of poverty areas and on the heterogeneity of the low-income groups (treating the poverty enclaves as ordinary spaces inhabited by ordinary citizens is also present in a scientific discussion). Additionally, the suggestion that the poor are less mobile so the proximity between wealthy and poor areas is an important factor also needs a comment as it is not always true.
Author Response
Point 1: The article presents a consistent and interesting research, with well defined aim and study area, and good theoretical background. I have only more general comment concerning the study’s assumptions. In the article, there is a suggestion that poverty is a cause of crime, which is supported by some theories. However, there is also a scientific discussion on stereotypes concerning enclaves of poverty, here mentioning only Wacquant’s poverty penalization, stigmatization of urban subploretariat and connected to them social and security policies. Deprived urban spaces treated as spaces of danger and violence with the poor treated as potential criminals is a rather simplistic approach. I understand the study presents only an excerpt of a research on social inequalities and on a geography of crime but in my opinion a comment is needed in theoretical background and in discussion sections on the complexity of poverty areas and on the heterogeneity of the low-income groups (treating the poverty enclaves as ordinary spaces inhabited by ordinary citizens is also present in a scientific discussion).
Response 1: Thank you so much for your feedback, you do raise some fair points. I have added a paragraph in the Discussion approaching the topic of assuming poverty is a motivator to crime, etc. The intention of the paper is to be somewhat neutral, not so much defending that localized inequality affects risk (which the empirical results did not support in the end!), but attempting to more thoroughly test an effect that has been implied by many criminological theories and that has often been taken for granted outside of academia. I mention this and a few other points in this new paragraph in the Discussion section.
Point 2: Additionally, the suggestion that the poor are less mobile so the proximity between wealthy and poor areas is an important factor also needs a comment as it is not always true.
Response 2: It is not my intention to suggest that the poor are less mobile! The assumption is that burglars (poor or not) would prefer closer targets, but also that poverty would be a motivator for burglary, leading then to the combined effect of nearby poverty boosting crime (which turned out to be not a significant effect). However, this mobility factor is an important point you raise. Hypothetically, even if the poor are to burglarize the rich (crudely speaking), if burglars have very high mobility they might not prefer closer locations, rendering localized income inequality irrelevant to the geography of burglary within cities. I have added a few sentences on that in the Discussion section.
Thank you again for your valuable feedback! I am open to provide further improvements if necessary.
Round 2
Reviewer 2 Report
The manuscript has been improved. I have no further comments.
Reviewer 3 Report
Very nice! Thank you for your edits.
Reviewer 4 Report
the text is to some extent revised accordingly to my comments as there is added a new possible interpretation of results in discussion section and a comment on the penalization of poverty in conclusions section. The revision is very basic however, in the article it is now suggested that the research problem is more complex.